# Allelopathic Potency and an Active Substance from *Anredera cordifolia* (Tenore) Steenis

**DOI:** 10.3390/plants8050134

**Published:** 2019-05-18

**Authors:** Ichsan Nurul Bari, Hisashi Kato-Noguchi, Arihiro Iwasaki, Kiyotake Suenaga

**Affiliations:** 1Department of Plant Pest and Disease, Faculty of Agriculture, Universitas Padjadjaran, Jalan Raya Bandung-Sumedang km 21, Sumedang, Jawa Barat 45363, Indonesia; 2Laboratory of Plant Biochemistry, Department of Applied Biological Science, Faculty of Agriculture, Kagawa University, Miki, Kagawa 761-0795, Japan; hisashi@ag.kagawa-u.ac.jp; 3Department of Chemistry, Faculty of Science and Technology, Keio University, 3-14-1 Hiyoshi, Kohoku, Yokohama 223-8522, Japan; a.iwasaki@chem.keio.ac.jp (A.I.); suenaga@chem.keio.ac.jp (K.S.)

**Keywords:** *Anredera cordifolia*, allelopathy, 3-hydroxy-alpha-ionone, seedling development

## Abstract

*Anredera cordifolia* (Tenore) Steenis is widely planted as an ornamental and medicinal plant in Indonesia. On the other hand, in some other countries this plant is classified as a noxious weed. As a harmful weed, *A. cordifolia* is reported to have the ability to smother all native vegetation, collapse canopies of tall trees, cultivate as a ground cover and disrupt native seedling development. There is no available information about the involvement of any allelochemicals from *A. cordifolia* related to these issues. The present study evaluated the allelopathic effect by isolating and identifying the allelopathic substance from *A. cordifolia* leaf extract. The allelopathic potency of *A. cordifolia* was determined by a series of bioassays of shoot and root growth on some selected test plants. Separation and purification of the active substances was achieved through several chromatography processes. Finally, the substances with allelopathic activity were identified through high-resolution electrospray ionization mass spectrometry (HRESIMS) analysis and determined by the specific rotation of compound, proton and carbon NMR spectroscopies. The results show that *A. cordifolia* possesses allelopathic properties which affect other plant species. The isolated compound from the plant material, 3-hydroxy-alpha-ionone, may contribute to the allelopathic effects of *A. cordifolia*.

## 1. Introduction

Madeira vine (*Anredera cordifolia* (Tenore) Steenis) is a succulent climbing vine belonging to the family Basellaceae. It is a very fast-growing plant which produces very long stems, up to 6 meters in length, as reproductive organs [1]. The Madeira vine also produces a mass of tubers as reproductive parts which are also a carbohydrate source that enables the plant to survive through difficult periods [2].

In Indonesia, the Madeira vine is recognized under the local name binahong, a famous folk medicinal plant, known especially for healing wounds and several diseases. The Encyclopedia of Traditional Chinese Medicines informs that *A. cordifolia* can be used to treat fractures, knocks, falls and weakness during convalescence; to disperse swelling and to dissipate stasis; as a supplement for kidney disease; to strengthen the lumbus and for relief of soreness in the lumbus and the knees [3]. The plant has also been used to treat diabetes, hepatitis and cardiovascular problems [4]. Some references also reported that the Madeira vine has other biological properties as an antioxidant [5], antihyperlipidemic agent, endothelial fat content reducer [6] and an antihyperuricemic agent [7]. Therefore, because of these known benefits, binahong is currently widely planted as an ornamental and medicinal plant in Indonesia.

On the other hand, the Madeira vine has become a noxious weed in some areas where it was introduced, including Australia [2], New Zealand [8], Hawai’i [9] and South Africa [10]. As a harmful weed, *A. cordifolia* has the ability to smother all native vegetation, collapse canopies of tall trees, cultivate as a ground cover and disrupt the development of native seedlings [11]. Some reports have noted that such abilities stem from the rapid and dense growth of the Madeira vine. Canopy collapse of mature trees is caused by the weight of the Madeira vine [2], while *A. cordifolia* may also cover the ground and inhibit the development of other sun-loving plant species [12]. Currently, there is no available information about the involvement of any allelochemicals from *A. cordifolia* related to these issues.

Numerous phyla of plants are reported to have allelopathic species. Allelopathy is a natural phenomenon of biochemical interactions among all types of plants, including microorganisms that influence the germination, growth, survival and reproduction of other species [13]. Therefore, this study aimed to assess allelopathic potentiality and to isolate and identify allelochemicals from *A. cordifolia*.

## 2. Materials and Methods

### 2.1. Plant Material

Madeira vine leaves were collected in August 2014, in Jatinangor-West Java, Indonesia (6°56′40.6″ S; 107°45′46.7″ E). The collected leaves were washed, air dried, powdered and kept at 2 °C until extraction.

### 2.2. Extraction

Powder from the plant material (400 g) was extracted with 2000 mL of 70% (v/v) aqueous methanol for 48 h. In the meantime, the crude extract was mixed using a spatula and filtrated using a sheet filter paper with ∅ 110 mm (No. 2 Toyo-Japan). The filtrate was extracted again for 24 h with 2000 mL of methanol, then both filtrates were merged and evaporated to dryness with a machine. The stocks of crude extracts were stored in 20 mL methanol and kept at 2 °C until the bioassay.

### 2.3. Bioassay of the Crude Extract and Determination of I_50_

Some test plants were used for the bioassay, including four dicots and four monocots. Alfalfa (*Medicago sativa* L.), garden cress (*Lepidium sativum* L.), lettuce (*Lactuca sativa* L.) and rapeseed (*Brassica napus*) were selected as the dicots, whereas barnyard grass (*Echinochloa crus-galli* (L.) Beauv.), foxtail fescue (*Vulpia myuros* (L.) C.C. Gmel.), Italian ryegrass (*Lolium multiflorum* Lam.) and timothy (*Phleum pratense* L.) were selected as the monocots. The test plants were chosen for the experiment because of their known seedling development behaviors and their common use as model plants for laboratory bioassays [14,15,16,17,18]. Germination percentages of the seeds were >90% on average.

The bioassay was conducted with six concentrations (1, 3, 10, 30, 100, 300 mg dry weight (DW) equivalent extract mL^−1^) and controls. An aliquot of the extracts was evaporated to dryness, kept in 2 mL methanol, dropped to a sheet of filter paper on Petri dishes (∅ 28 mm) and dehydrated inside the laminar flow. After drying, 0.6 mL of 0.05% (v/v) aqueous Tween 20 (Nacalai, Kyoto, Japan) solution was added to each filter paper with residue of the crude extract inside the Petri dishes. The controls were the same as the treatments, but without the leaf extracts. From every test plant, 10 seeds were set on the filter paper inside the Petri dishes. The Petri dishes were arranged in a tray, protected with plastic wrap and aluminum foil and kept for 48 h inside a growth chamber at a temperature of 25 °C. Percent inhibition of seedling development was determined by comparing the length of each treatment seedling to the length of the control seedling development, while the *I*_50_ values of each test plant species were calculated using logistic regression analysis between the concentration–response correlation. The *I*_50_ values were analyzed to establish the concentration needed to inhibit 50% growth of the seedlings.

### 2.4. Isolation

The rest of the crude extract stock was separated into ethyl acetate and the aqueous phase. Firstly, 100 mL of aqueous residue was fixed to pH 7.0 using 1 M phosphate buffer and partitioned three times using an equivalent volume of ethyl acetate. The ethyl acetate phase was used for the following isolation process. The sample was evaporated to dryness after using anhydrous Na_2_SO_4_ overnight. The obtained material was separated by a column of silica gel (60 g, silica gel 60, 70–230 mesh; Merck KGaA, Darmstadt, Germany). The active fraction (70% EtOAc in *n*-hexane) obtained from the silica gel column was dissolved in 20% (v/v) Aq. MeOH and further separated by a column of Sephadex LH-20 (100 g, Amersham Pharmacia Biotech, Buckinghamshire, UK). The column was eluted with 20%, 40%, 60% and 80% (v/v) Aq. MeOH (100 mL per step) and MeOH (200 mL). The active fraction (40% Aq. MeOH) obtained from the Sephadex LH-20 column was dissolved in 20% (v/v) Aq. MeOH and loaded into a reversed-phase C_18_ cartridge. The cartridge was eluted with 20%, 40%, 60% and 80% (v/v) Aq. MeOH (15 mL per step) and MeOH (20 mL). The active fraction (40% Aq. MeOH) obtained from the C_18_ cartridge was purified by reversed-phase HPLC. The column (YMC-Pack ODS-AQ (inner diameter: 5001 × 0 mm)) was set at room temperature. The wavelength of the UV detector was 220 nm. The flow rate was 1.5 mL/min. The mobile phase was 50% (v/v) Aq. MeOH. The inhibitory compound was eluted at a retention time of 83–88 min.

### 2.5. Identification of the Active Substance

The active compound was characterized by high-resolution electrospray ionization mass spectrometry (HRESIMS), determined by ^1^H (proton) and ^13^C (carbon) NMR spectra (CDCl_3_ as internal standard) and the specific rotation of the compound.

### 2.6. Bioassay of the Active Substance and Determination of I_50_

The allelopathic substance activity was determined using a series of bioassays with garden cress and barnyard grass as test plants. For the bioassays, the concentrations were 1, 3, 10, 30, 100, 300, 1000 µm and controls. The procedure followed was the same as the bioassay of the crude extract, detailed above.

### 2.7. Analysis

The trials were arranged in a completely randomized design (CRD). Each treatment of the bioassays of crude extracts had three replications and two repetitions, whereas each treatment of the bioassays of allelopathic substances had two replications. ANOVA and Duncan’s post hoc tests were analyzed by IBM^®^ SPSS^®^ Statistic Ver. 21.0 (for Macintosh) and the *I*_50_ values were determined by GraphPad Prism^®^ Ver. 6.0e (for Macintosh).

## 3. Results

### 3.1. Biological Activity of the Crude Extract

The biological activity of the *A. cordifolia* leaf extract on shoot and root growth is summarized in Figure 1. The results implied that each test plant species indicated different sensitivity to the extract of *A. cordifolia*. However, generally, the shoot and root growth was more inhibited when the concentration of the extract was added, in the case of both shoot and root growth. Dicots were more sensitive than monocots. Starting from the 100 mg DW equivalent extract mL^−1^, dicot species were completely inhibited, whereas monocot species were completely inhibited by the extracts starting from the 300 mg DW equivalent extract mL^−1^ (Duncan, *p* < 0.05). In the case of the barnyard grass and foxtail fescue, a different trend was observed compared to other test plant species at the lowest extract concentration, in that it was seen as a growth promoter.

The required concentration of *A. cordifolia* leaf extract for inhibiting 50% of seedling development (*I*_50_) is summarized in Table 1. The results show that amongst the dicot species, garden cress was the most sensitive test plant to the *A. cordifolia* leaf extract. It required 4.61 mg DW equivalent extract mL^−1^ to inhibit 50% of the shoot development and 3.45 mg DW equivalent extract mL^−1^ to inhibit 50% of the root development. On the other hand, amongst the monocot species, the most sensitive test plant species was the barnyard grass, in regard to root development, and timothy, in regard to shoot development. To inhibit 50% of the root development of barnyard grass, 7.12 mg DW equivalent extract mL^−1^ was required and to inhibit 50% of the shoot development of timothy, 11.19 mg DW equivalent extract mL^−1^ was required. The *I*_50_ values (Table 1) are analogous to the data from Figure 1, the results show that dicot species were more sensitive than monocot species. The averages of the *I*_50_ were 11.68 and 12.96 mg DW equivalent extract mL^−1^ for shoots and roots, respectively, whereas in the case of monocot species, the *I*_50_ averages were 27.03 and 25.00 mg DW equivalent extract mL^−1^ for shoots and roots, respectively.

### 3.2. Isolation and Identification of the Active Substance

The compound was a colorless, transparent residue with a sweet odor like violets and a woody, berry, floral taste, as described on the PubChem website [19]. The chromatogram of the compound appeared at 37–41 min retention time (Figure 2).

C_13_H_20_O_2_ was clarified as the molecular formula of the compound, which was characterized by HRESIMS at *m*/*z* 209.1530 [M + H]^+^ (calcd for C_13_H_21_O_2_, 209.1542, Δ = −1.2 mmu). The proton NMR spectrum (400 MHz, CDCl_3_ as internal standard) showed δ_H_ 6.53 (dd, *J* = 15.9, 10.3 Hz, I H, H-7), 6.10 (d, *J* = 15.9, I H, H-8), 5.63 (br s, I H, H-4), 4.27 (br s, I H, H-3), 2.50 (d, *J* = 10.3, I H, H-6), 2.26 (s, 3 H, H-10), 1.84 (dd, *J* = 13.9, 6.1, I H, H-2), 1.62 (d, *J* = 0.7, 3 H, H-13), 1.40 (dd, *J* = 13.9, 6.7, I H, H-2’), 1.03 (s, 3 H, H-11), 0.89 (s, 3 H, H-12). The carbon NMR spectrum (100 MHz, CDCl_3_ as internal standard) showed δ_c_ 198.3 (C-9), 147.3 (C-7), 135.6 (C-5), 133.8 (C-8), 125.9 (C-4), 65.6 (C-3), 54.4 (C-6), 44.0 (C-2), 34.0 (C-1), 29.4 (C-11), 27.3 (C-10), 24.8 (C-12), 22.8 (C-13). The specific rotation of the compound was ${\left[\propto \right]}_{D}^{25}+99$ (*c* 0.10, CH_2_Cl_2_). The active compound was identified as 3-hydroxy-alpha-ionone (Figure 3) by comparing the data with a previous report [20].

### 3.3. Biological Activity of the Active Substance

Bioassays of the 3-hydroxy-alpha-ionone on shoot and root growth were conducted with garden cress and barnyard grass as test plant species (Figure 4 and Figure 5, respectively). The results showed that the inhibition of shoot and root growth was amplified with the increase of the substance concentration, in both of the bioassays with garden cress and barnyard grass. A similar result was found in the biological activity of the *A. cordifolia* leaf crude extract in a bioassay of the 3-hydroxy-alpha-ionone. Dicot species were also more sensitive than monocot species. Garden cress seedling growth was completely inhibited from 100 µm of substance concentration, whereas barnyard grass seedling growth was completely inhibited from 300 µm of substance concentration (Duncan, *p* < 0.05).

*I*_50_ values of 3-hydroxy-alpha-ionone isolated from *A. cordifolia* leaf on shoot and root growth of garden cress and barnyard grass are summarized in Table 2. Results of the biological activity of *A. cordifolia* leaf crude extract, analogous to the calculations of the *I*_50_ value of 3-hydroxy-alpha-ionone, showed that garden cress, as a dicot species, was more sensitive than barnyard grass as a monocot species. For the inhibition of 50% of shoot and root growth in garden cress, 35.60 µm of substance concentration was required for the shoots and 38.03 µm of substance concentration for the roots, while for the inhibition of 50% of shoot and root growth in barnyard grass, 41.00 µm of substance concentration was required for the shoots and 53.24 µm of substance concentration for the roots.

## 4. Discussion

The allelopathic potency of *A. cordifolia* leaf extracts was determined by a series of bioassays on seedling development of selected test plant species, including alfalfa, garden cress, lettuce, rapeseed, barnyard grass, Italian ryegrass, foxtail fescue and timothy. The results confirmed that *A. cordifolia* leaves contain a naturally occurring allelopathic substance. In the field, the ability of *A. cordifolia* to smother native species, collapse canopies of mature trees and disrupt native seedling germination [11] may be caused by the involvement of chemical processes. In this case, *A. cordifolia* may release allelopathic substances to suppress and destroy other plant species. In nature, allelochemicals are likely released in four main ways, including through leaching, decomposition, volatilization and exudation [13]. The *I*_50_ values of the crude extract of *A. cordifolia* leaves indicated that the species has allelopathic effects on other plant species. Accordingly, *A. cordifolia* may become a harmful weed in the future, although currently in Indonesia and in other places, *A. cordifolia* is not yet defined as a weed.

For the utilization of *A. cordifolia*, with regard to weed management, a farmer can use the vines of *A. cordifoli* for various uses such as mulching or as a biological herbicide spray. This plant has been reported as a safe material to use for the environment and also for humans. The report noted that the extract of *A. cordifolia* leaf has no toxic signs or abnormalities. Thus, it can be considered safe for medicines and other uses [21], including for weed control as a biological herbicide.

In this study, the active substance, 3-hydroxy-alpha-ionone, was first reported as a compound isolated from the leaves of *A. cordifolia*. Ionone and its derivatives are an essential intermediate involved in terpenoid metabolism [22]. The substance has been isolated from raspberry fruits [23]. In this study, the ability of 3-hydroxy-alpha-ionone to inhibit the shoot and root growth of garden cress and barnyard grass was exposed. An analogous compound, 3-hydroxy-beta-ionone was isolated in previous research from *Rhynchostegium pallidifolium* moss [24], in which the compound was described as a growth inhibitor in a bioassay using garden cress as the test plant. For the inhibition of 50% of seedling development in garden cress bioassays, 35.60 µm of 3-hydroxy-alpha-ionone was required for shoots and 38.03 µm for roots and 14.90 µm of 3-hydroxy-beta-ionone was required for shoots and 16.3 µm for roots. We concluded that 3-hydroxy-beta-ionone may be stronger in activity than 3-hydroxy-alpha-ionone as a phytotoxic compound. Another ionone derivative from *Athyriurn yokoscense*, 3-hydroxy-5,6-epoxy-beta-ionone, was isolated and the compound was determined to be a growth inhibitor in lettuce seedlings [25].

Toxicological information of the compound, alpha-ionone, was summarized on the PubChem website. The compound caused some behavioral effects in rats and mice as target test organisms, with a lethal dose (LD_50_) value of 2277–2605 mg/kg body weight. The behavioral effects included somnolence or general signs of depression, epileptic seizures and sleepiness [19].

Alpha-Ionone is moderately soluble in water. If released in water or soil, it is expected to bind to soil particles or suspended particles [19]. The present results indicate that the isolated compound inhibited shoot and root growth of garden cress and barnyard grass. The mechanism of the inhibition could be caused by changes in the structure of plant cells, cell elongation inhibition, antioxidant system imbalances, the breakdown of activities and functions of various enzymes, the effects on nutrient absorption in plant roots or an influence on nucleic acid and protein synthesis [26]. In conclusion, *A. cordifolia* naturally has an allelopathic effect on other plant species. The isolated compound from the plant material, 3-hydroxy-alpha-ionone, may contribute to the allelopathic effects of *A. cordifolia*.

## Figures and Tables

**Figure 1 plants-08-00134-f001:**
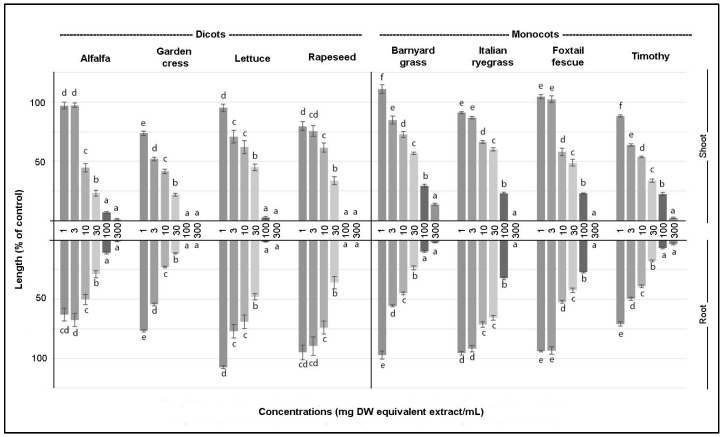
Biological activity of *Anredera cordifolia* leaf extract on the growth of shoots and roots of eight test plants. Different letters indicate significantly different values (Duncan, *p* < 0.05).

**Figure 2 plants-08-00134-f002:**
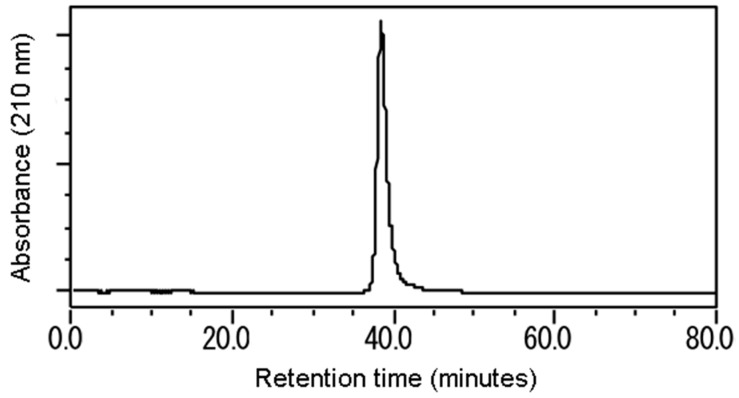
The chromatogram of the compound was checked by Waters HPLC (mobile phase: 40% Aq. MeOH, flow rate: 0.8 mL/min, inner diameter of column: 4.6 × 250 mm, S-5 µm), the retention time was 37–41 min, the optimum absorption wavelength was 210 nm.

**Figure 3 plants-08-00134-f003:**
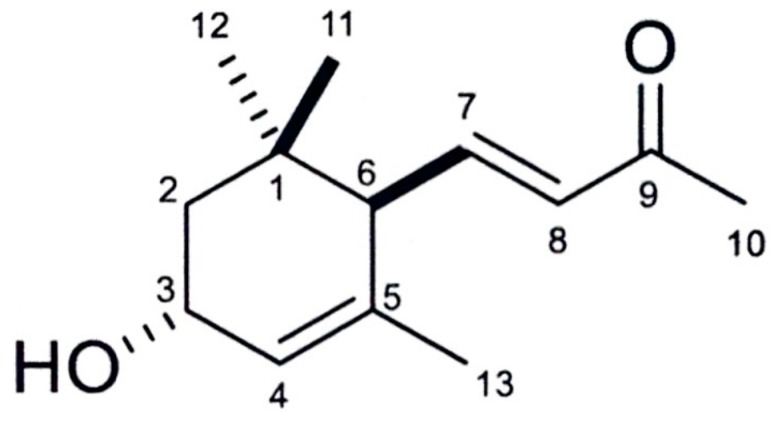
3-hydroxy-alpha-ionone structure.

**Figure 4 plants-08-00134-f004:**
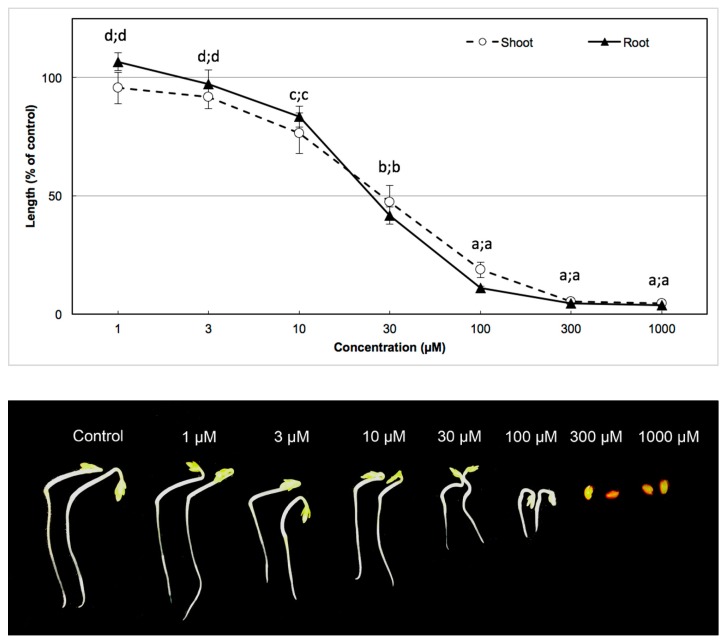
Biological activity of the compound 3-hydroxy-alpha-ionone on seedling growth of garden cress. Different letters indicate significantly different values (Duncan, *p* < 0.05).

**Figure 5 plants-08-00134-f005:**
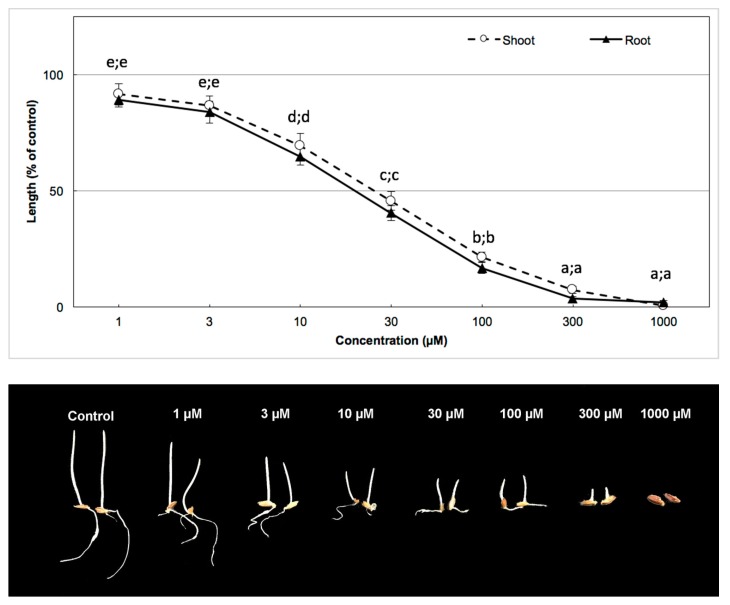
Activity of 3-hydroxy-alpha-ionone on seedling growth of barnyard grass. Different letters indicate significantly different values (Duncan, *p* < 0.05).

**Table 1 plants-08-00134-t001:** *I*_50_ of the aqueous methanol extract of *Anredera cordifolia* leaf on the shoot and root growth of eight plant species (mg dry weight (DW) equivalent extract mL^−1^).

Test Plants	Shoot	Root
**Dicots:**		
Alfalfa	11.00	6.43
Garden cress	4.61	3.45
Lettuce	16.26	21.17
Rapeseed	14.85	20.81
**Monocots:**		
Barnyard grass	38.31	7.12
Italian ryegrass	32.58	48.19
Foxtail fescue	26.03	20.98
Timothy	11.19	23.70

**Table 2 plants-08-00134-t002:** *I*_50_ of 3-hydroxy-alpha-ionone on shoot and root growth of garden cress and barnyard grass.

Test Plants	Shoot	Root
Garden cress	35.60 µm	38.03 µm
Barnyard grass	41.00 µm	53.24 µm

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
