# Peer review of "Allelopathic Potency and an Active Substance from Anredera cordifolia (Tenore) Steenis"

_plants, 2019, doi:10.3390/plants8050134_

Round 1

Reviewer 1 Report

The work is very preliminary. There is no indication about the mode of action or the ecological implications of the substance (i.e.: Is the substance volatile? Could induce decay of other plants when the leaf is on ground? Is it produced in other plant organs?

The discussion section is very poor.

Author Response

Dear Respected Reviewer,

Thank you very much for giving me your valuable time to review my manuscript entitled "Allelopathic potency and an active substance from Anredera cordifolia (Tenore) Steenis"

As your suggestion, some part of the manuscript have been improved especially at the section of Introduction and Discussion. I have added some references related to your questions. I am so sorry if it is not sufficient, especially about your last question "Is it produced in other plant organs?". I cannot answer the question due to it need further investigation. No available information, because it is the first study about allelopathy of A. cordifolia. I have noted it for my consideration in further research. Thank you.

Reviewer 2 Report

Review of the manuscript “Allelopathic potency and an active substance from Anredera cordifolia (Tenore) Steenis” by Ichsan Nurul Bari, Hisashi Kato-Noguchi, Arihiro Iwasaki, Kiyotake Suenaga

General comments

The manuscript investigates the allelopathic effects of Anredera cordifolia leaf extract on shoot and root growth of four dicots and four monocots.The manuscript provides interesting indications on an environmental-friend approach in the field of weed management and control highlighting the potential application of A. cordifolia as bio herbicide.

My major criticism regards the Introduction and Discussion sections which need to be implemented because too short. I suggest that this ms should be accepted after some changes given below.

Abstract

·         L 18. What do you mean with “the role of allelopathic mechanism”. In my opinion, the present study evaluated the allelopathic effects of Anredera cordifolia leaf extract, but not its role. Please explain it better.

Introduction

20 lines are really little for an Introduction. Literature is plenty of studies about allelopathy or Anredera cordifolia, therefore I strongly suggest implementing your introduction section supplying a more complete state-of-the-art.

Materials and Methods

·         L53-54. Change “inside a fridge (+ 2oC)” to “at 4 °C”

·         L 60. Change “inside refrigerator” to “at 4 °C”

·         L 66-68. “The test plants were chosen for the experiment because of their known seedling development behaviors and their commonly being used as a model plant for laboratory bioassay.” Please supply some references about that.

·         L 143-146. “Allelopathic substance of A. cordifolia leaf extract was separated and purified through (…) were determined by bioassay with garden cress as test plant.” This is already reported in Materials and Methods, do not repeat it in Result section.

Discussion

As for the Introduction, the Discussion is quite poor. For example, literature is plenty of studies about ionone, therefore it should not be so difficult to perform a more substantial discussion.

·         L202-204. “A. cordifolia may release allelopathic substances to suppress and destroy other plants species. The I50 values of crude extract of A. cordifolia leaf indicated that the species has allelopathic effects to other plants species.” To be precise, you investigated the allelopathic effects of Anredera cordifolia leaf extract, but you do not know if those substances in the extract are released in nature by Anredera cordifolia. You do not know if A. cordifolia can actually release that substances when it still alive. Please take into account this consideration when comment your results in Discussion section.

·         L223. I guess it should be “and THE compound was”.

Author Response

Dear Respected Reviewer,

Thank you very much for giving me your valuable time to review my manuscript entitled "Allelopathic potency and an active substance from Anredera cordifolia (Tenore) Steenis"

As your suggestion, some part of the manuscript have been improved especially at the section of Introduction and Discussion. I have added some references related to your questions. I did my best, but I am so sorry if it is not sufficient. 

The detail of my response as follows:

General comments

The manuscript investigates the allelopathic effects of Anredera cordifolia leaf extract on shoot and root growth of four dicots and four monocots.The manuscript provides interesting indications on an environmental-friend approach in the field of weed management and control highlighting the potential application of A. cordifolia as bio herbicide.

My major criticism regards the Introduction and Discussion sections which need to be implemented because too short. I suggest that this ms should be accepted after some changes given below.

Abstract

L18. What do you mean with “the role of allelopathic mechanism”. In my opinion, the present study evaluated the allelopathic effects of Anredera cordifolia leaf extract, but not its role. Please explain it better.

My response:

The sentence was modified to "The present study evaluated the allelopathic effect" as your suggestion. Thank you.

Introduction

20 lines are really little for an Introduction. Literature is plenty of studies about allelopathy or Anredera cordifolia, therefore I strongly suggest implementing your introduction section supplying a more complete state-of-the-art.

My response:

The section of Introduction has been improved by adding some related references as your suggestion. Thank you.

Materials and Methods

L53-54. Change “inside a fridge (+ 2oC)” to “at 4 °C”

My response:

The sentence was modified to "at 2°C" . Thank you.

L60. Change “inside refrigerator” to “at 4 °C”

My response:

The sentence was modified to "at 2°C" . Thank you.

L66-68. “The test plants were chosen for the experiment because of their known seedling development behaviors and their commonly being used as a model plant for laboratory bioassay.” Please supply some references about that.

My response:

Some references were supplied as your suggestion. Thank you.

L143-146. “Allelopathic substance of A. cordifolia leaf extract was separated and purified through (…) were determined by bioassay with garden cress as test plant.” This is already reported in Materials and Methods, do not repeat it in Result section.

My response:

The sentences were deleted as your suggestion. Thank you.

Discussion

As for the Introduction, the Discussion is quite poor. For example, literature is plenty of studies about ionone, therefore it should not be so difficult to perform a more substantial discussion.

L202-204. “A. cordifolia may release allelopathic substances to suppress and destroy other plants species. The I50 values of crude extract of A. cordifolia leaf indicated that the species has allelopathic effects to other plants species.” To be precise, you investigated the allelopathic effects of Anredera cordifolia leaf extract, but you do not know if those substances in the extract are released in nature by Anredera cordifolia. You do not know if A. cordifolia can actually release that substances when it still alive. Please take into account this consideration when comment your results in Discussion section.

My response:

Yes, you are right. In the present study we just investigated the allelopathic effect of Anredera cordifolia leaf extract as you mention. In nature, allelochemicals are possibly released by four main ways, such as leaching, decomposition, volatilization and exudation (Reference was add in the manuscript). I just want to highlight the involvement of chemical compound in the phenomenon of the ability of A. cordifolia to smother native species, collapse canopy of mature trees and disrupt native seedling germination. About "how the substance are released by the plant?" it is need further investigation. I have noted it for my consideration in further research. Thank you.

L223. I guess it should be “and THE compound was”.

My response:

The word "he" was modified to "the" as your suggestion. Thank you.

With my best,

Ichsan Nurul Bari, Ph.D.

Plants EISSN 2223-7747 Published by MDPI AG, Basel, Switzerland RSS E-Mail Table of Contents Alert
Back to Top